# SANS: Efficient Densest Subgraph Discovery over Relational Graphs without Materialization

## ABSTRACT

How can we efficiently identify the densest subgraph over relational graphs? Existing dense subgraph discovery (DSD) approaches assume that a relational graph $H$ is already derived from a heterogeneous data source and they focus on efficient discovery of the densest subgraph on the materialized $H$. Unfortunately, materializing relational graphs can be resource-intensive, which thus limits the practical usefulness of existing algorithms over large datasets. To mitigate this, we propose a novel Summary-bAsed deNsest Subgraph discovery (SANS) system. Our unique *summary-based peeling* algorithm forms the core of SANS. Following the peeling paradigm, it utilizes summaries of each node's neighborhood to efficiently estimate peeling coefficients and subgraph densities at each peeling iteration and thus avoids materializing the relational graph completely. Through extensive experiments, we demonstrate the efficacy and efficiency of SANS, reaching orders of magnitude speedups compared to the conventional baselines with materialization, while consistently achieving at least 95% accuracy compared to peeling algorithms based on materialization.

## 1 INTRODUCTION

Discovering dense subgraphs in a given network has broad applications in different domains such as system optimization by social piggybacking [12], discovery of protein complexes [26, 32], information dissemination analysis to discover filter bubbles and echo chambers [18, 19, 21], and forms a building block of many graph problems including reachability queries [8]. Most prior studies on densest subgraph discovery assume that a materialized data graph is available and accessible with low latency. However, in many applications, the underlying network on which to discover densest subgraphs is not explicitly available and tends to be present implicitly in the form of complex relationships between entities induced by meta-paths [9, 22, 28, 35]. Materializing relational graphs, either offline or online, can be prohibitively expensive.

Company A (anonymized for double-blinded review), one of the leading technology enterprises, provides various online services such as digital payment transactions and food delivery and collects and stores relational data in heterogeneous format. To support business needs, they often perform analyses on relational graphs derived from the heterogeneous data using meta-paths. For example, market segmentation relies on community detection over a relational graph induced by the "(customer)→ (merchant)", while fraud detection over relational graphs induced by meta-path "(customer)→(merchant) → (delivery-person)" is used to detect potential collusion between delivery persons and deal-hunters. Materializing these relational graphs online can be both time and memory-prohibitive. Specifically, our experiments reveal that relational graph construction accounts for up to 99.58% of the total time needed for discovering dense subgraphs over relational graphs. Additionally, businesses such as market segmentation often require on-demand analysis over different time spans to track the changes in customer preferences, making pre-materialization across all time spans impractical.

To this end, in this work, we address the challenge of *densest subgraph discovery* (DSD) within relational graphs shaped by a specific meta-path $\mathcal{M}$, while avoiding the need to materialize them.

**Our Contributions.** To address these challenges, we propose a novel system, Summary-bAsed deNsest Subgraph discovery (SANS) for DSD. Our system distinguishes itself from prior studies by introducing the following technical contributions.

*Summary-based Peeling (Sec. 3).* The core component of SANS is the summary-based peeling algorithm. Specifically, instead of obtaining the complete set of neighborhoods of each node, SANS *directly estimates* densities utilizing neighborhood summaries of each node in the relational graph, which can be constructed efficiently from heterogeneous data sources through meta-path supervised summary propagation. This strategic approach sidesteps the extensive resource demands typically associated with the materialization of relational graphs. Combining with lazy maintenance of the neighborhood summaries that becomes outdated due to removal of nodes during peeling iterations, SANS showcases orders of magnitude of superiority in terms of time and space efficiency over contemporary methods for detecting the densest subgraph based on materialized relational graphs.

*Theoretical Results (Sec. 4).* We prove that SANS provides a $(2 + \epsilon)$-approximation to the corresponding DSD problem based on edge-density metric. For triangle density, we devise a novel unbiased estimator for the peeling coefficient, i.e., the number of triangles containing each node and the triangle density for subgraphs generated during peeling iterations. Furthermore, our summary-based peeling algorithm can support any density metric that can be computed by the cardinality of node neighborhoods or cardinality of set operations (union, intersection or difference supported by neighborhood summaries [4]) of node neighborhoods.

*Empirical Results (Sec. 5).* Experiments on real-world datasets demonstrate that the SANS system scales well to large relational graphs where baselines fail to terminate in a reasonable time and achieves orders of magnitude speedup in most cases over baselines, while yielding subgraphs with at least 95% density and comparable size to the subgraph returned by peeling algorithms based on materialization. We also deploy SANS for fraudulent account detection over the heterogeneous data from company A. The results demonstrate that SANS can identify frauds with precision over 97%.

## 2 PRELIMINARIES

### 2.1 Notations and Problem Formulation

For concreteness of exposition, we model a heterogeneous data source as a knowledge graph (KG). Specifically, a KG is denoted

as $\mathcal{G} = (\mathcal{V}, \mathcal{E}, \mathcal{L})$, where $\mathcal{V}$ and $\mathcal{E}$ represent the sets of nodes and edges. An edge $e \in \mathcal{E}$ connects two nodes $u, v \in \mathcal{V}$. The function $\mathcal{L}$ maps each node or edge, $v$ or $e$, to its type, $\mathcal{L}(v)$ or $\mathcal{L}(e)$. For simplicity of exposition, we assume that $\mathcal{L}(e)$ is solely determined by the node types of it end nodes. Nevertheless, our methods can directly handle multiple edge types between a pair of node types. *Note that our formulation and methods are orthogonal to the format of the heterogeneous data and can be extended to support other data sources such as relational databases.*

*Meta-paths*, originally introduced in [28], are commonly used to extract relational graphs from KGs [9, 17, 25, 35, 37].

DEFINITION 1 (META-PATH). *An L-hop meta-path is a sequence of node types denoted as* $\mathcal{M} = (x_0, x_1, \ldots, x_{L-1}, x_L)$, *where* $x_i$ *is the type of the i-th node. The inverse of* $\mathcal{M}$ *is denoted as* $\mathcal{M}^{-1} = (x_L, x_{L-1}, \ldots, x_1, x_0)$.

Although our methods support any meta-path $\mathcal{M}$, following the established convention [9, 15, 35], we consider symmetric meta-paths which induces homogeneous relational graphs. A meta-path $\mathcal{M}$ is symmetric if $\mathcal{M} = \mathcal{M}^{-1}$.

DEFINITION 2 (MATCHING INSTANCE). *A matching instance M of a meta-path* $\mathcal{M}$, *denoted* $M \triangleright \mathcal{M}$, *is a sequence of nodes* $(v_0, v_1, \ldots, v_{L-1}, v_L)$ *in* $\mathcal{G}$ *satisfying:*
- $\forall i \in \{0, \ldots, L\}, \mathcal{L}(v_i) = x_i$.
- $\forall i \in \{0, \ldots, L-1\}, (v_i, v_{i+1}) \in \mathcal{E}$.

*We use* $M(x_i)$ *to denote node* $v_i$ *in instance M, of the node type* $x_i$.

DEFINITION 3 (RELATIONAL GRAPH). *For a given KG* $\mathcal{G}$, *a symmetric meta-path* $\mathcal{M}$ *induces a relational graph* $H_\mathcal{M} = (V_\mathcal{M}, E_\mathcal{M})$ *from* $\mathcal{G}$ *where:*
- $V_\mathcal{M}$ *contains all nodes in* $\mathcal{V}$ *that match* $x_0$ *in any instance of* $\mathcal{M}$;
- $E_\mathcal{M}$ *contains all edges* $e = (u, v)$ *for any two nodes* $u, v \in V_\mathcal{M}$ *such that there are* $M \triangleright \mathcal{M}$ *in* $\mathcal{G}$ *with* $M(x_0) = u$ *and* $M(x_L) = v$.

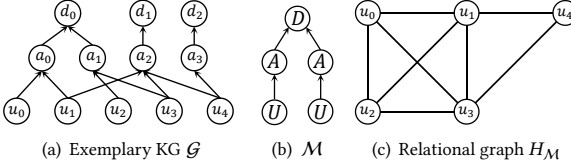

| (a) Exemplary KG $\mathcal{G}$ | (b) $\mathcal{M}$ | (c) Relational graph $H_\mathcal{M}$ |

**Figure 1: A relational graph $H_\mathcal{M}$ derived from a KG $\mathcal{G}$ using a symmetric meta-path $\mathcal{M}$.**

EXAMPLE 1. *Fig. 1 presents an example e-commerce KG with three node types U ("user"), A ("account"), D ("device"). Sequence (U, A, D, A, U) denotes a symmetric meta-path* $\mathcal{M}$, *which induces the relational graph* $H_\mathcal{M}$ *in Fig. 1(c). For instance, two nodes* $u_0$ *and* $u_2$ *are neighbors in* $H_\mathcal{M}$ *because there exist an instance* $M = (u_0, a_0, d_0, a_1, u_2)$ *of* $\mathcal{M}$. *More intuitively, two users* $u_0$ *and* $u_2$ *are connected since they ever login through the same device* $d_0$.

We denote the set of neighbors of $u$ in $H_\mathcal{M}$ as $\mathcal{N}(u)$ and the *degree* of $u$ in $H_\mathcal{M}$ as $N(u) = |\mathcal{N}(u)|$. Furthermore, $\Lambda = \max_{u \in H} N(u)$ denotes the maximum node degree in $H_\mathcal{M}$. When the context is clear, we drop the subscript $\mathcal{M}$ and denote a relational graph as $H = (V, E)$ for simplicity.

**Induced Subgraph.** Let $H = (V, E)$ be a relational graph and $S \subseteq V$. The induced subgraph of $H$ *w.r.t.* $S$ is the subgraph $H[S] = (S, E[S])$,

where $E[S] = \{(u, v) \in E \mid u \in S \land v \in S\}$. In the rest of the paper, we abuse the notation $S$ to also denote the subgraph it induces.

**Density Metrics** $\rho(\cdot)$. We define a density metric $\rho$ for a subset $S$ of vertices as $\rho(S) = \frac{w(S)}{|S|}$, where $w(S)$ represents the weight of the induced subgraph $H[S]$. Our proposed system supports a variety of density metrics, including those where $w(S)$ represents the number of edges (edge-density) [13], number of triangles ($\Delta$-density) [29], or more generic functions w.r.t. node degrees [31] within $H[S]$.

For example, we give the definitions of edge- and $\Delta$-density.

DEFINITION 4 (EDGE-DENSITY). *Given an induced subgraph* $H[S] = (S, E[S])$, *its edge-density is defined as* $\rho_e(S) = \frac{|E[S]|}{|S|}$.

DEFINITION 5 ($\Delta$-DENSITY). *For an induced subgraph* $H[S] = (S, E[S])$, *its triangle-density, also denoted as* $\Delta$-density, *is* $\rho_\Delta(S) = \frac{\Delta(S)}{|S|}$, *where* $\Delta(S)$ *is the number of triangles in* $H[S]$.

The problem of *densest subgraph discovery* (DSD) over relational graphs is formally defined as follows.

PROBLEM 1 (DSD). *Given a relational graph H (not necessarily materialized) and a density metric* $\rho(\cdot)$, *find a set of vertices* $S^* \subseteq V$ *such that* $\rho(S^*)$ *is maximized i.e.,* $S^* = \arg\max_{S \subseteq V} \rho(S)$.

For example, in Fig. 1(c), the subgraph induced by the node set $S^* = \{u_0, u_1, u_2, u_3, u_4\}$ forms the densest subgraph with density 1.6 over the relational graph $H$ in terms of the edge-density metric.

In this work, we focus on discovering the set of nodes $S^*$ forming the densest subgraph instead of returning the induced subgraph $H[S^*]$. Note that after finding the set of nodes $S^*$, based on application requirements, $H[S^*]$ can be materialized quite efficiently by disregarding nodes outside of $S^*$. Exact algorithms for DSD require time-consuming max-flow computations. In this work, we follow the efficient peeling paradigm, which returns an approximated densest subgraph with theoretical guarantees.

**Peeling Coefficient** $\varphi_S(u)$. The peeling coefficient function, denoted by $\varphi_S(u)$, quantifies a node's marginal contribution to the density of the induced subgraph $H[S]$. Its calculation is dependent on the chosen density metric. We define $\varphi_S(u)$ as follows:

$$\varphi_S(u) = w(S) - w(S \setminus \{u\}) \tag{1}$$

When the context is clear, we drop $S$ and denote the peeling coefficient as $\varphi(u)$ for simplicity.

**Peeling Paradigm.** The peeling paradigm is a general iterative approach for DSD. Given a relational graph $H$, it begins by computing the initial peeling coefficient of each node in $H$ and then obtains a sequence of subgraphs by peeling nodes iteratively. In each iteration, it peels the node with minimum *peeling coefficient*, updates the peeling coefficient for the set of nodes whose peeling coefficient is influenced by the peeling, and maintains the subgraph with largest density. Finally, the subgraph of the largest density is returned. The *peeling coefficient* is determined by the density metric to achieve good approximations of the optimal. Our work adopts the peeling paradigm because it can efficiently recover approximate densest subgraphs with constant approximation guarantees while avoiding expensive max-flow or linear programming computations required by exact approaches, as discussed in the related work below.

## 2.2 Related Work

**Relational Graph Materialization.** A straightforward solution of our problem is to first materialize the relational graph and then execute DSD algorithms over it. Instead of enumerating all instances of meta-paths, there has been a line of works on improving the efficiency for relational graph materialization. For example, Chatzopoulos et al. [7] proposed a method to enumerate instances of different meta-paths through workload sharing. Guo et al. [14] proposed the algorithm to materialize relational graphs by boolean matrix multiplication, specially optimized for graphs with locally dense regions. The generic worst-case optimal join [1, 11, 24, 30] algorithm can also be used for relational graph materialization through efficient traversal of meta-path instances avoiding duplicate visit of nodes in multiple meta-path instances. Our experiments (Sec. 5.2) reveal that the materialization costs remain time-consuming and are the bottleneck for DSD in relational graphs.

**Densest Subgraph Discovery.** There is a rich literature on the problem of DSD. Exact DSD algorithms involve solving a max-flow [10, 13] or linear programming [6] problem, which is not scalable to large graphs. Thus, the peeling paradigm, which is efficient with approximation guarantees, has been widely adopted in the literature. Charikar [6] proposed the peeling algorithm for edge-density, which iteratively removes nodes with smallest degree and returns the subgraph with largest edge-density generated during the peeling iterations. Bahmani et al. [2] proposed to remove all nodes with degree less than $2(1 + \epsilon)\rho$ at each peeling iteration and prove that this algorithm achieves a solution of $2(1 + \epsilon)$-approximation. Boob et al. [5] proposed the multi-round peeling algorithm, which obtains the densest subgraph by iteratively removing the node with the smallest load, where the load of a node in each round is the sum of its induced degree and its load in previous round. The peeling paradigm has also been extended to other density metrics. Tsourakakis et al. [29] modeled the graph density as the number of $k$-cliques contained in the graph and showed that the peeling algorithm of repeatedly removing the node contained in the smallest number of corresponding cliques achieves $k$-approximation.

Besides, graph reduction based optimizations [10, 34, 39] have been explored for efficient densest subgraph discovery, which restricts expensive max-flow or linear programming computations within a subgraph containing the densest subgraph.

*All these works assume that the relational graph is already materialized and thus cannot scale to large heterogeneous data sources owing to the requirement of materializing the relational graph. Specifically, graph reduction-based optimizations still require relational graphs to be fully materialized before reduction.*

**Community Search over Heterogeneous Information Networks (HIN).** Recently, several works have focused on the community search problem over HINs based on meta-paths.

Fang et al. [9] proposed to reveal $(k, \mathcal{M})$-core from HINs, which is equivalent to the $k$-core in the relational graph induced by meta-path $\mathcal{M}$. The algorithm dynamically maintains $k$ neighbors for each node in the relational graph through meta-path instances and removes nodes with less than $k$ neighbors iteratively. Yang et al. [35] introduced the $(k, \mathcal{M})$-Btruss and $(k, \mathcal{M})$-Ctruss models for community search in KGs. The $(k, \mathcal{M})$-Btruss is equivalent to the $k$-truss in the relational graph induced by $\mathcal{M}$, whereas $(k, \mathcal{M})$-Ctruss considers the overlaps between meta-path instances and does not correspond to existing models of cohesive subgraphs in the relational graphs. Jiang et al. [15] studied the nested meta-path core $(k, \Psi)$-core, which aims at finding a subgraph that is $(k, \mathcal{M})$-core for each meta-path $\mathcal{M} \in \Psi$.

The above methods focus on discovering subgraphs conforming to certain topological community models with strict topological constraints and can overlook relevant subgraphs with complex topological structures [36]. Instead, our work focuses on finding the subgraph optimizing the density metric, which is more flexible than topology-driven community search methods.

Strouthopoulos and Papadopoulos [27] studied the discovery of $k$-cores in hidden networks, where the existence of any edge can only be decided via a *probe operation*. However, this work also aims to find subgraphs that conform to the $k$-core model. Besides, their method cannot handle large relational graphs since the probe operations are time-consuming enumerations of meta-paths.

## 3 THE SANS SYSTEM

SANS is motivated by the observation that only the *peeling coefficient* of each node and the *densities* of the sequence of generated subgraphs during peeling iterations, which can be computed based on statistics such as degree or number of triangles within the neighborhood rather than the exact set of neighbors for each node in the relational graph, are required by the peeling paradigm. Specifically, SANS avoids relational graph materialization by constructing and maintaining neighborhood summaries for each node in the relational graph, which are capable of peeling coefficient and subgraph density estimation during peeling iterations. *Our system further provides user defined APIs for peeling coefficient and subgraph density estimation to streamline its deployment for various density metrics (described in Appendix A.1 due to space limit).*

Intuitively, the neighborhood summary is a $k$-sized subset of each node's neighbors, uniformly sampled from its complete neighborhood in the relational graph.

**Summary Construction.** The neighborhood of each node in $H$ can be obtained by propagating nodes in the *matching graph*.

DEFINITION 6 (MATCHING GRAPH). *Given a KG $\mathcal{G}$ and a meta-path $\mathcal{M} = (x_0, x_1, \ldots, x_{L-1}, x_L)$, the matching graph of $\mathcal{M}$ over $\mathcal{G}$ is a multi-level graph $\mathcal{G}^*(\mathcal{V}^*, \mathcal{E}^*)$ with node set $\mathcal{V}^* = \bigcup_{0 \le i \le L} \mathcal{V}_i$ and edge set $\mathcal{E}^* = \bigcup_{0 \le i < L} \mathcal{E}_i$, where $\mathcal{V}_i$ is the set of nodes of type $x_i$ contained in any matching instance of $\mathcal{M}$ and $\mathcal{E}_i$ consists of all edges that connect nodes between $\mathcal{V}_i$ and $\mathcal{V}_{i+1}$.*

Intuitively, each path from a node in $\mathcal{V}_0$ to $\mathcal{V}_L$ represents an instance of the query meta-path $\mathcal{M}$. By propagating all nodes in $\mathcal{V}_0$, along the edges of the matching graph from $\mathcal{V}_0$ to $\mathcal{V}_L$, each node in $\mathcal{V}_L$ receives all its neighbors in $H$.[1] However, explicitly obtaining all neighbors of each node leads to a prohibitive time complexity of $O(|V| \cdot |\mathcal{E}^*|)$ since each node in $\mathcal{V}^*$ potentially retains all the nodes it has received and then propagates these nodes to its successors through the edges in $\mathcal{G}^*$.

To avoid such costs, we generate a random number uniformly between interval $(0, 1)$ for each node $v \in V$, denoted $r(v)$. To begin

---

[1]Note that both $\mathcal{V}_0$ and $\mathcal{V}_L$ are equal to the node set $V$ of relational graph $H$.

with, each node in $\mathcal{V}_0$ maintains itself and the random number generated for it. During the propagation, each node in $\mathcal{V}_{i-1}$ propagates the nodes and random numbers it maintains to its successors and each node in $\mathcal{V}_i$ will only retain nodes corresponding to $k$-smallest random numbers it receives. Finally, each node in $\mathcal{V}_L$ will receive $k$ neighbors along with the corresponding random numbers forming the summary of its neighborhood. Since the random numbers are generated uniformly for each node in the relational graph and each neighborhood summary contains the $k$ neighbors corresponding to the smallest $k$ random numbers, the $k$ neighbors in the neighborhood summary of a given node forms a uniformly sampled subset of its neighborhood. Combined with the random number generated with each node, our neighborhood summary is a cardinality sketch [3, 4], which can be used to estimate a variety of peeling coefficients for DSD. As such, we circumvent the prohibitive costs of materializing the relational graph.

To achieve good approximation for peeling coefficients and subgraph densities, we perform multiple ($\theta$) independent propagations and construct $\theta$ different neighborhood summaries for each node.

**Coefficients estimation.** In general, our summary-based peeling algorithm can support any density metric that can be computed by the cardinality of node neighborhoods or cardinality of set operations (union, intersection or difference supported by neighborhood summaries [4]) of node neighborhoods. We describe edge-density (cardinality of node neighborhoods) and $\Delta$-density (intersection of node neighborhoods) in detail as two representative metrics. We also discuss the estimator for $p$-mean density [31] and density metrics with general size functions [16] in Appendix A.2. Expectations in estimators are implemented by taking average over the $\theta$ independent summaries for each node.

*Edge-density.* The peeling coefficient of node $v$, i.e. its degree $N(v)$, and the density of subgraph $S$ generated during peeling iterations are estimated by estimators $\tilde{N}(v)$ and $\tilde{\rho}_e(S)$ respectively [3, 4]

$$\tilde{N}(v) = \frac{k}{\mathbb{E}[\kappa(v)]} - 1 \qquad \tilde{\rho}_e(S) = \frac{\sum_{v \in S} \tilde{N}(v)}{2} \qquad (2)$$

where $\kappa(v)$ denotes the $k$-th smallest random number $v$ receives.

*$\Delta$-density.* To estimate the peeling coefficients $\Delta(v)$, we note that $\Delta(v) = \frac{\sum_{u \in N(v)} \Delta(u,v)}{2}$, where $\Delta(u,v)$ denote the number of triangles containing edge $(u,v)$ and can be estimated through neighborhood summaries of $u$ and $v$ as

$$\tilde{\Delta}(u,v) = \tilde{N}(v) + \tilde{N}(u) - \left( \frac{k}{\mathbb{E}[\kappa(u,v)]} - 1 \right)$$

where $\kappa(u,v)$ denote the $k$-th smallest distinct random number among the union of random numbers received $u$ or $v$.

Thus, we estimate $\Delta(v)$ as the multiplication of degree $N(v)$ and the average number of triangles containing an uniformly sampled adjacent edge of $v$. To ensure the unbiasedness of the estimator, we divide $\theta$ summaries of each node into 3 partitions, and use each partition to estimate $N(v)$, $\Delta(u,v)$ and to sample adjacent edge of $v$ respectively.

$$\tilde{\Delta}(v) = \tilde{N}(v) \cdot \frac{\sum_{u \in \mathcal{K}_v} \tilde{\Delta}(u,v)}{2 \cdot |\mathcal{K}_v|} \qquad \tilde{\rho}_\Delta(S) = \frac{\sum_{v \in S} \tilde{\Delta}(v)}{3} \qquad (3)$$

where $\mathcal{K}_v$ denote the set of uniformly sampled neighbors contained in the partition of summaries for adjacent edge sampling.

EXAMPLE 2. *Fig. 2(a) presents a running example of neighborhood summary construction through propagation over the matching graph.*

*For clarity, we only present the propagation process of summary construction for node $u_0$. To begin with, we generate a random number for each node in $\mathcal{V}_0$, and then propagate nodes in $\mathcal{V}_0$ along with corresponding random numbers to $u_0$ in $\mathcal{V}_4$. During the propagation, each node in the matching graph will only keep the 2 nodes corresponding to the smallest random numbers it receives. Specifically, $d_0$ receives four nodes $\{u_2, u_1, u_3, u_0\}$ from $a_0$ and $a_1$, but it will only keep two nodes with 2-smallest random numbers. When propagation has finished, $u_0$ in $\mathcal{V}_4$ receives its neighborhood summaries $\{(u_2, 0.2), (u_1, 0.4)\}$. Neighborhood summaries of other nodes are constructed similarly and depicted in Fig. 2(b), where they are utilized for coefficients estimation.*

*Suppose the edge-density metric is considered, the peeling coefficient of $u_4$ is estimated as $\tilde{N}(u_4) = \frac{k}{\kappa(u_4)} - 1 = 2.33$, which has smallest estimated degree. Thus, node $u_4$ is going to be removed in the first peeling iteration. The density of the current remaining subgraph is estimated as $\tilde{\rho}_e(S_0) = \sum_{i=0}^{4} \tilde{N}(u_i) = 1.83$.*

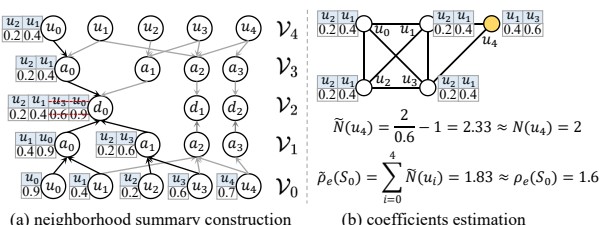

$$\tilde{N}(u_4) = \frac{2}{0.6} - 1 = 2.33 \approx N(u_4) = 2$$

$$\tilde{\rho}_e(S_0) = \sum_{i=0}^{4} \tilde{N}(u_i) = 1.83 \approx \rho_e(S_0) = 1.6$$

(a) neighborhood summary construction  (b) coefficients estimation

**Figure 2: Neighborhood summary construction (subfigure (a)) and coefficient estimation (subfigure (b)) based on edge-density metric and $k = 2$. Subfigure (a) illustrates the detailed propagation for neighborhood-summary construction of $u_0$, where elements with red strikethrough are discarded due to $k$-size limitation. Note that the relational graph in (b) is only for demonstration and not materialized in our algorithm.**

**Summary-based Peeling with Lazy Maintenance.** Summary-based peeling algorithm employs neighborhood summaries for DSD over relational graphs efficiently without materialization. To begin with, the algorithm first constructs neighborhood summaries through propagation over the matching graph. In each peeling iteration, the algorithm estimates the peeling coefficients for each node based on neighborhood summaries and removes the node with minimum estimated peeling coefficients. After node peeling, the algorithm estimates the density of the remained subgraph. The algorithm proceeds until only one node is remained in the relational graph. Finally, the subgraph generated during peeling iterations with largest estimated density is returned. The detailed pseudo-code is deferred to Appendix A.3 due to space limit.

During peeling iterations, nodes are iteratively removed and the remaining subgraph is changed continuously. Thus, the neighborhood summaries can become outdated. To avoid reconstruct neighborhood summaries at each peeling iteration, we propose a *lazy summary maintenance* approach for efficient maintenance of neighborhood summaries. At each peeling iteration, when a node $v$ is peeled, we remove $v$ and the corresponding random number $r(v)$ from neighborhood summaries containing $v$. Since each original neighborhood summary before maintenance retains neighbors corresponding to the $k$ smallest random numbers, after removing $v$, it still retains the $k-1$ neighbors corresponding to smallest

random numbers, which can still be used for peeling coefficient and subgraph density estimation with slightly reduced accuracy. We reconstruct neighborhood summaries whenever their size falls below a given threshold $k^-$ to strike a balance between the number of reconstructions and estimation accuracy.

## 4 THEORETICAL ANALYSIS

In this section, we prove that the SANS system based on neighborhood summaries provides good estimators for DSD with different density metrics. Specifically, for DSD over relational graphs based on edge density, we prove that SANS can return a subgraph with constant approximated edge density compared to the optimal densest subgraph. For DSD based on $\Delta$-density, we prove that SANS provides an unbiased estimation of peeling coefficients and $\Delta$-densities of subgraphs in each peeling iteration. The detailed all proofs are deferred to Appendix A.4 due to space limit.

**Edge-density.** Charikar [6] shows that by using degree $N(v)$ of each node $v$ as the *peeling coefficient*, the peeling algorithm, run on a materialized relational graph, can achieve a 2-approximation guarantee for the densest subgraph based on edge-density. We extend Charikar's result and prove that the summary-based algorithm gives a $2(1+\epsilon)$-approximation of DSD with high probability (w.h.p.).

THEOREM 1. *Let $S$ denote the subgraph returned by SANS, we have $\rho_e(S) \geq \frac{\rho_e(S^*)}{2(1+\epsilon)}$ w.p. at least $(1-p)$, provided $\theta = \Theta(\frac{\Lambda}{\epsilon k} \log \frac{|V|}{p})$.*

$\Delta$-**density.** We note that the intersection operation required by estimating number of triangles breaks the probabilistic bounds of our estimators. We can nevertheless prove that the estimator $\tilde{\Delta}(v)$ and $\tilde{\rho}_\Delta(S)$ based on neighborhood summaries are unbiased estimators for the peeling coefficient $\Delta(v)$ and density of any subgraph $S$ generated during peeling iterations.

THEOREM 2. *Given any subset $S \subseteq V$ and $v \in S$, we have $\Delta(v) = \mathbb{E}[\tilde{\Delta}(v)]$ and $\rho_\Delta(S) = \mathbb{E}[\tilde{\rho}_\Delta(S)]$.*

Further, we give the analysis over the complexity of the summary-based peeling algorithm for different density metrics.

THEOREM 3. *The time complexity of summary-based peeling is $O(k(\theta + \theta^*)(|\mathcal{E}^*| + |V|f(k) + |V|\log|V|))$ and the space complexity is $O(k\theta|\mathcal{V}^*| + |\mathcal{E}^*|)$, where $f(k)$ denotes the time complexity for peeling coefficient estimation using neighborhood summaries and $\theta^*$ denotes the total number of neighborhood summaries reconstructed during the peeling iterations.*

The time complexity is significantly influenced by the number $\theta$ of summaries constructed as well as the number $\theta^*$ of summaries reconstructed due to node removals during the peeling iterations. The value of $\theta$ is theoretically bounded in the worst case so that our method achieves approximation guarantees for edge-density and triangle density. The value of $\theta^*$, which is influenced by both the specific structure of the relational graph as well as the density metric, is often small as indicated by our experiments. We leave the theoretical study of $\theta^*$ as a future work. Function $f(k)$ is determined by the target density metric.

We then analyze in detail the complexity of our SANS system when deploying it to edge-density and $\Delta$-density respectively.

**Edge-density.** The time complexity of edge-density optimization with the SANS system is $O(k(\theta + \theta^*)(|\mathcal{E}^*| + |V|k + |V|\log|V|))$

**Table 1: Statistics of KGs used in the experiments, where $|\mathcal{L}(\mathcal{V})|$ and $|\mathcal{L}(\mathcal{E})|$ denote the numbers of node and edge types, and $|\mathbb{M}|$ is the number of evaluated meta-paths.**

| Dataset | $|\mathcal{V}|$ | $|\mathcal{E}|$ | $|\mathcal{L}(\mathcal{V})|$ | $|\mathcal{L}(\mathcal{E})|$ | $|\mathbb{M}|$ |
|---------|------|------|------|------|------|
| IMDB | 21.4K | 86.6K | 4 | 6 | 679 |
| ACM | 10.9K | 548K | 4 | 8 | 20 |
| DBLP | 26.1K | 239.6K | 4 | 6 | 48 |
| PubMed | 63.1K | 236.5K | 4 | 10 | 172 |
| FreeBase | 180K | 1.06M | 8 | 36 | 151 |

according to Theorem 3 since peeling coefficient estimation requires $f(k) = O(k)$ time to find the largest random number in the neighborhood summary. In practice, we find that $\theta^*$ is small when optimizing the edge-density metric (see Sec. 5.2). The intuition behind this is the following. The neighborhood summary of a node will only be updated during those peeling iterations when the removed node happens to correspond to a random number in the neighborhood summary. For a node $v$ with large degree, the update of its neighborhood summary will only take place with a small probability $k/N(v)$ due to its large neighborhood. On the other hand, for nodes with small degrees, since nodes are removed in the ascending order of degrees, these nodes will be removed during early peeling iterations and thus also do not require summary reconstruction w.h.p.

$\Delta$-**density.** The time complexity of triangle-density DSD with SANS is $O(k(\theta + \theta^*)(|\mathcal{E}^*| + |V|k^2 + |V|\log|V|))$ since peeling coefficient estimation has complexity $f(k) = O(k^2)$. Specifically, computing the intersection of two summaries has complexity $O(k)$, and $k$ intersections are required since $k$ neighbors are sampled in the summary. We empirically observed (see Table 2 in Sec. 5.2) that $\theta^*$ is small when optimizing triangle-density. Although nodes are removed in ascending order w.r.t. the number of triangles containing them instead of degree, we can still expect a small $\theta^*$ in practice since the triangle number is positively correlated with degree.

## 5 EXPERIMENTS

### 5.1 Experimental Setup

**Datasets.** We conduct our experiments on five real-world datasets consisting of KGs . The statistics of the datasets are reported in Table 1. IMDB is a KG about actors and directors of movies. ACM and DBLP are two academic KGs denoting the co-authorships between researchers through papers and publishing venues. PubMed is a biomedical KG representing the relationships between genes, chemicals, and diseases of different species. FreeBase is extracted from a general-purpose KG developed by Google.

**Meta-path Generation.** We use AnyBURL [20] to extract a set of candidate meta-paths $\mathbb{M}$. We set the confidence threshold to 0.1 for AnyBURL and take the snapshot at ten seconds. Note that our proposed approaches are orthogonal to the method for meta-path generation. Column $|\mathbb{M}|$ in Table 1 shows the number of meta-paths found on each dataset.

**Compared Methods.** To our best knowledge, no prior works have focused on DSD over *unmaterialized* relational graphs. We thus compare peeling algorithms implemented through our SANS system with algorithms based on explicitly materialized relational

graph. For fairness of comparison, we wanted to give the baselines the advantage of the most efficient materialization. We experimented with various approaches for relational graph materialization including matrix multiplication [14] and Leapfrog TrieJoin [30] (more details in Appendix A.5). We found Leapfrog TrieJoin to be the most efficient and used it for materializing the relational graph for the baselines.[2]

- FlowE [13] and FlowT [23] apply the max-flow method to find the optimal densest subgraph given the materialized relational graph in terms of edge-density and $\Delta$-density respectively.
- CoreExact [39] is a core-based exact algorithm for DSD in terms of edge-density.
- KCCA [38] is a counting-based approach for exact DSD in terms of $\Delta$-density.
- PeelE [6] and PeelT [29] apply the peeling paradigm to find the approximate densest subgraph given a materialized relational graph, in terms of edge-density and $\Delta$-density respectively.
- SansE and SansT are peeling algorithms implemented through SANS in terms of edge-density and $\Delta$-density respectively.

Fang et al. [9] propose to use edge- and vertex-disjoint core models for community search over meta-path induced relational graphs. While effective for certain meta-paths, this model is not suitable for general meta-paths with extra node constraints due to its rigid topological restriction. For example, in IMDB, it will only return an edge/vertex-disjoint 2-core for 28.7% and 26.8% meta-paths discovered by AnyBURL respectively. Thus, we do not compare with these models.

**Parameter Setting.** In our experiments, we set $k = 24$, $k^- = 4$ and $\theta = 1$ for both SansE and SansT by default. We conduct parameter sensitivity analysis of $k$ and $\theta$ for efficiency and effectiveness in Sec. 5.2 and Sec. 5.3 respectively, and choose the default values of hyperparameters to ensure that SansE and SansT return a subgraph with a density larger than 0.95 of the densities achieved by methods based on materialized relational graphs on average across datasets, while minimizing the time consumption. Results in Sec. 5.3 demonstrate that peeling algorithms based on our SANS system with default parameters can in practice yield results comparable to materialization-based peeling methods.

**Evaluation Metrics.** (1) For efficiency, we report the average runtime of each algorithm to find the densest subgraph per meta-path. (2) For effectiveness, we quantify by the ratio $\gamma$ between the density of the subgraph returned by our methods (SansE and SansT) and the density achieved by the corresponding materialization-based peeling methods (PeelE and PeelT).

**Environment.** All experiments are conducted on a Linux Server with AMD EPYC 7643 CPU and 256 GB memory running Ubuntu 20.04. All algorithms are implemented in C++ with -O3 optimization and executed in a single thread. We release the source code of SANS in an anonymous repository [3].

## 5.2 Efficiency Evaluation

**Time Consumption.** In Table 2, we present the execution time per meta-path for each method. We have the following observations.

*First*, our methods SansE and SansT based on the SANS system are much more efficient than materialization-based methods.

---

[2]Leapfrog TrieJoin is also known to be worst-case optimal.
[3]https://anonymous.4open.science/r/SANS-CFFC/README.md

Materialization-based methods cannot handle all experimental cases in reasonable time. FlowE and FlowT can only handle small datasets: FlowE does not terminate in 24 hours except on datasets IMDB and ACM, while FlowT only completes on IMDB, due to time-consuming global flow computations. CoreExact is more efficient than FlowE as it performs flow computation only on dense cores. Nevertheless, it still cannot handle datasets PubMed and FreeBase within 24 hours due to time-consuming flow computations. Materialization-based methods KCCA and PeelT for $\Delta$-density do not complete on DBLP, PubMed, and FreeBase due to the time consuming triangle counting or enumeration operation. In contrast, our methods can handle all experimental cases efficiently. SansE and SansT complete on the largest dataset FreeBase within 0.34 and 3.64 seconds per meta-path respectively on average. Compared to materialization-based peeling methods, SansE achieves up to ~53× speedup over PeelE on PubMed and FreeBase, and SansT achieves up to 1063× speedup over PeelT on ACM.

*Second*, we can observe that methods based on SANS system are more robust to complex density metrics than prior peeling algorithms based on materialization, in terms of efficiency. To see this, notice that for PeelE optimizing edge-density metric, the relational graph materialization stage dominates its time consumption. The materialization time consistently occupies more than 85% of total time of PeelE across datasets and up to 99.58% on dataset ACM. However, this ratio drops significantly for PeelT optimizing $\Delta$-density, which is much more time-consuming than PeelE. Specifically, on dataset ACM, PeelE finds the densest subgraph based on edge-density within 3.0141 seconds, whereas the corresponding method PeelT requires 743.11 seconds for finding the densest subgraph based on $\Delta$-density on average due to time consuming triangle enumeration. On the other hand, SansT can still find densest subgraphs based on $\Delta$-density efficiently. The reason is that leveraging the neighborhood summaries of limited size, SansT can restrict the time-consuming computations of peeling coefficients and thus the running time becomes less sensitive to the density metrics.

Overall, SANS accelerates DSD by simultaneously avoiding the high computational overhead of materialization and also expediting the time-consuming peeling coefficient computation.

**Memory Usage.** Fig. 3 compares the memory usage of the materialized relational graph required by PeelE and PeelT with the neighborhood summaries utilized by SansE and SansT. For each dataset, we report the peak memory usage among all meta-paths. We have the following observations.

First, the additional memory usage required by the neighborhood summaries is consistently smaller than that of the materialized relational graphs across all datasets. Furthermore, the gap between the memory usage of neighborhood summaries and materialized relational graphs increases with the increase in the sizes of datasets. Specifically, on small datasets IMDB, neighborhood summaries requires 7.7× less memory than the the materialized relational graphs, whereas on the large dataset FreeBase, neighborhood summaries require up to 471× less additional memory than materializing the relational graphs explicitly. The reason is that the memory usage of neighborhood summaries is linear in the number of nodes in the dataset, whereas the materialized relational graph can consume up to the square of number of nodes in the dataset in the worst case.

**Table 2: Runtime of different algorithms. For algorithms based on materialization, the returning time of each dataset is reported. For SansE and SansT, we additionally report the speedup ratios over PeelE and PeelT, and the number of reconstructed summaries $\theta^*$. We also report the time consumption for relational graph materialization in the last column. Cells with a dash line denote that the method cannot be finished within 24 hours.**

| Metric | $\rho_e$ | | | | | | $\rho_\Delta$ | | | | | | Materialize |
|---|---|---|---|---|---|---|---|---|---|---|---|---|---|
| Method | FlowE | CoreExact | PeelE | SansE | | | FlowT | KCCA | PeelT | SansT | | | |
| | | | | Time | Speedup | $\theta^*$ | | | | Time | Speedup | $\theta^*$ | |
| IMDB | 0.0017s | 0.0018s | 0.0013s | 0.0008s | **1.53×** ⇑ | 1.24 | 0.1430s | 0.0570s | 0.0020s | 0.0018s | **1.10×** ⇑ | 1.24 | 0.0011s |
| ACM | 31.6030s | 19.5700s | 3.0141s | 0.2400s | **12.60×** ⇑ | 4.75 | - | 2779.63s | 723.11s | 0.6800s | **1063×** ⇑ | 4.65 | 3.0014s |
| DBLP | - | 624.87s | 4.8328s | 0.1554s | **31.00×** ⇑ | 10.02 | - | - | - | 1.5500s | - | 10.48 | 4.7200s |
| PubMed | - | - | 4.5380s | 0.0850s | **53.40×** ⇑ | 5.22 | - | - | - | 1.0100s | - | 5.20 | 4.4173s |
| FreeBase | - | - | 18.0464s | 0.3400s | **53.10×** ⇑ | 8.27 | - | - | - | 3.6400s | - | 9.32 | 17.4360s |

Scalability test (Appendix A.5) over synthetic data generated from TPCH benchmark demonstrate that SANS can handle KGs with upto 29.6M nodes.

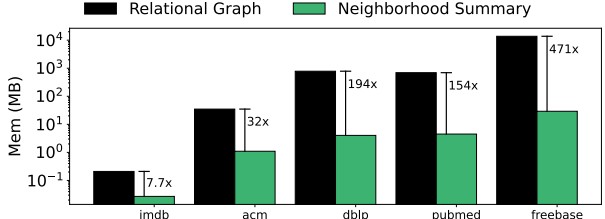

**Figure 3: Memory consumption of relational graphs materialized in PeelE/PeelT, and neighborhood summaries constructed in SansE/SansT.**

**Parameter Analysis.** We further study the influence of the neighborhood summary parameters $k$ and $\theta$ over the efficiency of our algorithms. Fig. 4(a)-4(e) reports the running time of SansE while varying $k$ and $\theta$ across datasets. We can observe that with the increase of $k$, the running time of SansE decreases monotonically across all datasets except ACM and PubMed, on which the time of SansE first decreases when $k$ is varied from 8 to 12 (resp. 16) and then increases (resp. slightly increases) with further increase of $k$. The reason is that with the increase of $k$, the number of neighborhood summaries to be reconstructed $\theta^*$ decreases and thus makes the algorithm less time-consuming (detail in Appendix A.5). However, when the value $k$ reaches a break point (e.g., > 12 on ACM and > 16 on PubMed), the time for neighborhood summaries construction and peeling coefficient estimation based on neighborhood summaries becomes the dominant factor and thus the running time starts to increase with $k$. For the other datasets, it is conceivable that the break point will be reached at larger values of $k$, i.e., $k > 24$.

On the other hand, with the increase of $\theta$, the running time of SansE increases significantly. The reason is that the larger value of $\theta$ increases the time for summary construction and peeling coefficient estimation and increases the number of summaries reconstructed $\theta^*$ during the peeling iterations.

Fig. 4(f)– 4(j) reports the running time of SansT while varying $k$ and $\theta$ across datasets. The running time of SansT increases monotonically with $k$ across datasets except FreeBase. The reason is that the time needed for peeling coefficient computation for $\Delta$-density metric is influenced by $k$ more significantly. As discussed in Sec. 4, the time complexity of method SansT is $O(k^3)$ due to the estimation of triangles containing each node, compared with $O(k^2)$ for SansE. Thus, the cost for peeling coefficient estimation starts to dominate the running time of SansT even for relatively small values of $k$ and it only increases with the increase of $k$.

Due to the space limit, we leave the experiments for varying parameter $k^-$ and asymptotic analysis of $\theta^*$ to our Appendix A.5.

## 5.3 Effective Evaluation

For gauging effectiveness, we compare the density of the subgraph returned by SansE (resp. SansT) with that returned by PeelE (resp. PeelT), the materialization based peeling algorithm. Fig. 5 reports the average ratio $\gamma$ between the edge-density of subgraph returned by SansE and the edge-density of subgraph returned by PeelE across datasets while varying parameters $k$ and $\theta$.

We first observe that the value $\gamma$ increases monotonically with both $k$ and $\theta$ across all datasets and approaches 1, which is consistent with Theorem 1 that SansE provides a $(2 + \epsilon)$-approximation to the DSD problem based on edge-density $\rho_e$.

Furthermore, the effectiveness of SansE is affected by the summary size $k$ more significantly than by the number of summaries $\theta$. Thus, in order to achieve better effectiveness, users may prefer to increase the summary size $k$ instead of increasing the number of summaries $\theta$. For example, in FreeBase, increasing $k$ from 8 to 16, fixing $\theta = 1$, boosts $\gamma$ from 90% to 95%, whereas increasing $\theta$ from 4 to 8, fixing $k = 8$, boosts $\gamma$ from 95% to 97%. Following this observation, we choose the default values of $k = 24$ and $\theta = 1$, which ensures that SansE achieves $\gamma > 95\%$ across all datasets. For SansT, we only compare the effectiveness of SansT with PeelT on IMDB and ACM, the two datasets on which PeelT finishes in reasonable time. We find that SansT returns subgraphs corresponding to a $\gamma$ of 99.65% and 95.7% on IMDB and ACM respectively.

Overall, by setting the default value of $k = 24$ and $\theta = 1$, both SansE and SansT are able to achieve 95%-approximation to the density of the subgraphs returned by the corresponding materialization-based peeling methods. Further, our experiments in Appendix A.5 reveal that SansE and SansT return densest subgraphs with sizes comparable to those returned by PeelE and PeelT.

## 5.4 Case Study

To further gauge the effectiveness of SANS over real-world applications, we apply SansE and SansT on heterogeneous data obtained from our anonymous industrial collaborator company A. We obtain three different types of nodes – *account*(A), *merchant*(M), *device*(D) and two types of edges account→merchant ("order"), account→device ("login"). We choose the meta-path "(account) → (device)" which induces a relational graph connecting two accounts if they login through the same device. Such a relational graph tends to form dense subgraphs containing fraudulent accounts. For each account in the dataset, it contains a ground-truth label denoting whether this account is a fraudster. In scenarios such as artificially

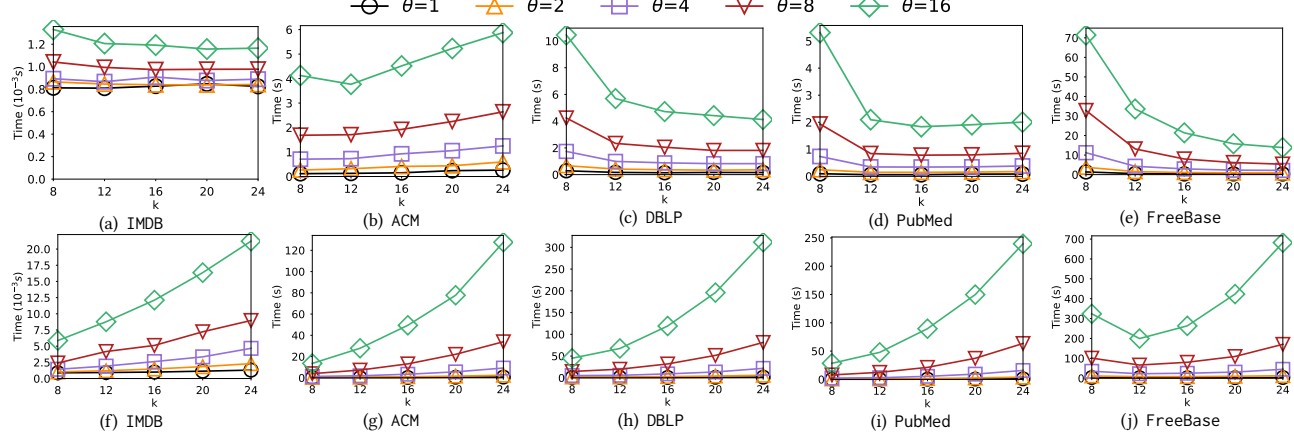

Figure 4: Efficiency analysis of SansE and SansT varying $k$ and $\theta$. Subfigures (a)-(e) reports the running time of SansE; subfigures (f)-(j) reports the running time of SansT.

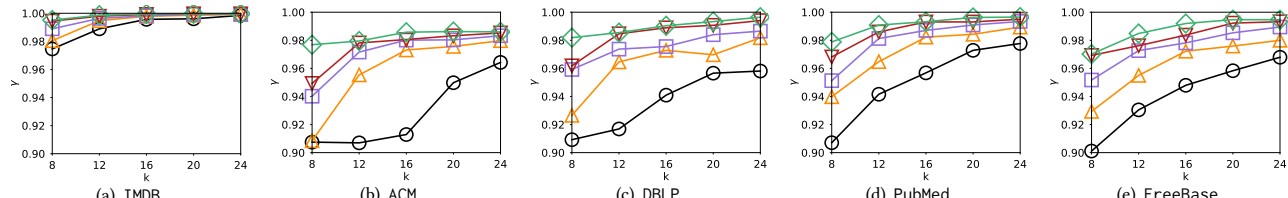

Figure 5: Effectiveness analysis of SansE with varying parameters $k$ and $\theta$ (Share legend with Fig. 4).

inflating a merchant's ranking and visibility through bulk ordering, fraudsters typically create numerous accounts to post positive reviews. These fraudulent actions often lead to the creation of a significant number of accounts, each logging in once, which results in a sparse representation in the heterogeneous graph, making many fraudulent accounts undetectable by DSD applied on the heterogeneous graph directly. However, as long as these accounts are connected through shared devices, they will form dense subgraphs on the relational graph, thus finding the densest subgraph on the relational graph is more promising for finding the fraudulent accounts.

Figure 6 illustrates the fraudulent community identified by our method SansE (Fig. 6(a)) and a normal community (Fig. 6(b)) formed through device sharing between family members. Traditional DSD methods cannot distinguish the fraudulent community from the normal one within the heterogeneous graph. Specifically, the normal community has density 1.875 in the original heterogeneous data (Fig. 6(b) left), which is even larger than the density of fraudulent community (Fig. 6(a) left). On the other hand, by finding the communities on the relational graphs induced by "(account) $\rightarrow$ (device)", the density of the fraudulent community (Fig. 6(a) right) increases to 38.05, which is significantly larger than the corresponding density of the normal community (Fig. 6(b) right). Of the 135 accounts flagged by SansE, 132 are genuinely fraudulent accounts, with 3 being normal accounts (false positives), resulting in a precision of 97.8%. Similarly, the $\Delta$-density-based method, SansT, identifies the same fraudulent community as SansE, along with an additional component comprising 9 fraudulent accounts, thus achieving a slightly higher precision of 98.6%.

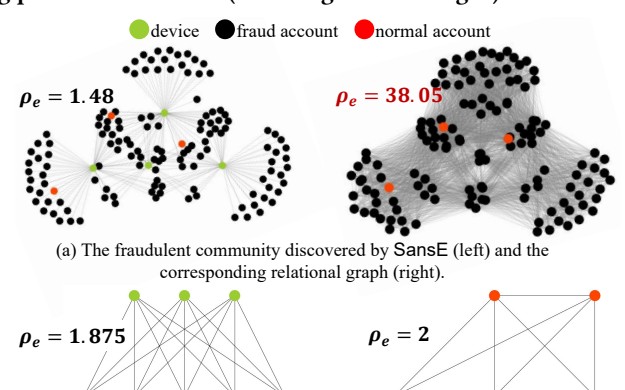

(a) The fraudulent community discovered by SansE (left) and the corresponding relational graph (right).

(b) A normal community (left) and its corresponding relational graph (right)

Figure 6: Fraudulent community returned by SansE (Subfigure (a)) and a normal community (Subfigure (b)).

## 6 CONCLUSION

In this paper we introduce SANS, a materialization-free system for DSD over relational graphs. Our system, grounded in the *summary-based peeling* approach, facilitates efficient estimation of peeling coefficients and subgraph density during peeling iterations directly from heterogeneous data sources. We establish that SANS can achieve constant approximation guarantees for DSD based on both edge- and $\Delta$-density. Extensive experiments show that peeling algorithm based on the SANS system are significantly more efficient than baselines across various datasets based on both edge- and $\Delta$-density, while returning subgraphs whose density is over 95% that of the subgraphs found by traditional materialization-based peeling algorithms.

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

# A APPENDIX

## A.1 APIs in SANS

Crafting the complete SANS system for different density metrics can be cumbersome for data engineers, given the significant effort required to construct and maintain the summaries within the system. SANS streamlines this endeavor and provides a set of interfaces to efficiently implement peeling algorithms for different density metrics. This allows data engineers to concentrate on selecting the best density metric and the corresponding peeling coefficient for their particular real-world application.

**Application Programming Interfaces (APIs).** Listing 1 presents the key APIs in SANS system through which users can deploy

```
1045  1 class SANS{
1046  2    virtual double coefficient(int v, summarySet K);
1047  3    virtual double density(vector<int> S, summarySet K);
1048  4    // S represents the subgraph induced by nodes in it
1049  5    double val(summary k);
1050  6 }
```

**Listing 1: APIs of SANS**

the SANS system for DSD based on various density metrics over relational graphs. Parameter summarySet K contains the set of summaries corresponding to each node in $V$, whereas summary k denotes one specific summary within K.

- coefficient() estimates the *peeling coefficient* for node v utilizing summaries in K. This API is called when computing the initial peeling coefficients as well as updating the peeling coefficient of influenced nodes after each removal during peeling iterations.
- density() estimates the density of the given subgraph S utilizing the summaries of nodes in S according to density metrics implemented by the user. This API is called to estimate the density of the sequence of subgraphs generated during peeling iterations.
- val() gives the estimation of the cardinality of the collection represented by the summary k. This API can be used by users for implementing the APIs coefficient() and density().

**Implementations for $\rho_e(\cdot)$.** Class EdgeSANS in Listing 2 implements the corresponding peeling algorithm in the SANS system. The API coefficient(v, K) performs function val on the neighborhood summary of node v to obtain an estimation of its degree (Line 3). For each subgraph S generated during peeling iterations, the API density(S, K) estimates its edge-density as the average degree of each node in S divided by 2 (Lines 6-8).

```
1079  1 class EdgeSANS: public SANS{
1080  2    double coefficient(int v, summarySet K) {
1081  3        return val(K[v]);
1082  4    }
1083  5    double density(vector<int> S, summarySet K){
1084  6        double den;
1085  7        for(int v: S) den += coefficient(v, K);
1086  8        return den / (S.size() * 2);
1087  9    }
      10 }
```

**Listing 2: Edge-densest Subgraph Discovery in SANS**

**Implementation of $\rho_\Delta(\cdot)$.** Listing 3 gives the implementation of the SANS system for triangle-densest subgraph discovery. The operator & (Line 4) is reloaded as the intersection between two summaries. The function coefficient(v, K) first estimates the degree of node $v$ (Line 3), and then for each neighbor of $v$ sampled within the summary K[v], it counts the number of triangles containing edge $(u, v)$ based on summary intersections and records the total number of triangles in variable t (Line 4). Finally, the estimator is computed according to Eq. 3 and returned (Line 5). The API density(S, K) is implemented to estimate the triangle-density of a given subgraph S by calling coefficient for each node in S.

```
1103  1 class TriangleSANS: public SANS{
1104  2    double coefficient(int v, summarySet K) {
1105  3        double d = val(K[v]), t = 0, s = 0;
1106  4        for(int u: K[v].nbr()) t += val(K[u] & K[v]);
1107  5        return (t * d) / (K[v].size() * 2);
1108  6    }
1109  7    double density(vector<int> S, summarySet K) {
1110  8        double t = 0;
1111  9        for(int v: S) t += coefficient(v, K);
1112  10        return t / (S.size() * 3);
       11    }
       12 }
```

**Listing 3: Triangle-densest Subgraph Discovery in SANS**

## A.2 SANS for Additional Density Metrics

In this section, we further briefly describe how to deploy SANS to other density metrics including $p$-mean density $\rho_p(S)$ and density metrics with convex/concave size functions $\rho_g(S)$.

$p$-**mean density** $\rho_p(S)$ was first proposed by Veldt et al. [31] as a generalization of edge-density.

DEFINITION 7 ($p$-MEAN DENSITY). *Given a graph $S = (V_S, E_S)$ and parameter $p \in \mathbb{R}$, the $p$-mean density of $S$ is defined as $\rho_p(S) = (\frac{1}{|V_S|} \sum_{v \in S} N_S(v)^p)^{1/p}$.*

According to the definition of the peeling coefficient in Eq. 1, we can obtain the corresponding peeling coefficient for $p$-mean density as:

$$\varphi(u) = N(u)^p + \sum_{v \in \mathcal{N}(u)} (N(v)^p - (N(v) - 1)^p) \qquad (4)$$

Veldt et al. [31] proved that for any $p \geq 1$, the peeling algorithm based on the above peeling coefficient can yield a $(1 + p)^{1/p}$-approximation densest subgraph in terms of $p$-mean density.

To deploy our SANS system on $p$-mean density metric, we provide the following estimator for $\varphi_S(u)$ based on neighborhood summaries:

$$\tilde{\varphi}(u) = \tilde{N}(u)^p + \frac{\tilde{N}(u) \cdot \sum_{v \in \mathcal{K}_u} (\tilde{N}(v)^p - (\tilde{N}(v) - 1)^p)}{|\mathcal{K}_u|} \qquad (5)$$

The estimator for the $p$-mean density of each subgraph S generated during the peeling iterations can be defined as:

$$\tilde{\rho}_p(S) = (\frac{1}{|V_S|} \sum_{v \in S} \tilde{N}_S(v)^p)^{1/p} \qquad (6)$$

List 4 gives the implementation of the SANS system for $p$-mean densest subgraph discovery based on the peeling coefficient estimator in Eq. 5 and density estimator in Eq. 6.

**Density metric $\rho_g(S)$ with general size function $g(S)$.** We further discuss how to deploy SANS system for the optimization of density metrics with general size functions, which generalize the edge-density in another direction.

DEFINITION 8. *Given a subgraph $S = (V_S, E_S)$ and a monotonically non-decreasing function $g : \mathbb{N}^+ \to \mathbb{R}$ with $g(0) = 0$, the size function based density based on $g$ of $S$ is defined as $\rho_g(S) = \frac{|E_S|}{g(|V_S|)}$.*

Kawase and Miyauchi [16] proved that when the function $g$ is either convex or concave, the peeling algorithm with degree as peeling coefficient can be used for $\rho_g(S)$ optimization with certain

```
1  class PmeanSANS: public SANS{
2      double coefficient(int v, summarySet K) {
3          double d = val(K[v]), t = 0;
4          for(int u: K[v].nbr())
5              t += (pow(val(u), p) - pow(val(u)-1, p));
6          return pow(d, p) + d * t / K[v].size();
7      }
8      double density(vector<int> S, summarySet K){
9          double density;
10         for(int v: S) density += pow(val(K[v]), p);
11         return pow(density / S.size(), 1 / p);
12     }
13 }
```

**Listing 4: $p$-mean densest Subgraph Discovery in SANS**

approximation guarantees. Class GeneralSizeSANS in Listing 5 implements the corresponding peeling algorithm in the SANS system. The API coefficient(v, K) returns the estimated degree of node v. For each subgraph S, the API density(S, K) estimates its density as the summation of the degree of each node in S divided by 2 times $g(|V_S|)$.

```
1  class GeneralSizeSANS: public SANS{
2      double coefficient(int v, summarySet K) {
3          return val(K[v]);
4      }
5      double density(vector<int> S, summarySet K){
6          double den;
7          for(int v: S) den += coefficient(v, K);
8          return den / 2 * g(S.size());
9      }
10 }
```

**Listing 5: Densest Subgraph Discovery in SANS with generalized size function**

### A.3 Pseudo-code

Alg. 1 gives the pseudocode for the summary-peeling algorithm. It receives a KG $\mathcal{G}$, a meta-path $\mathcal{M}$, the number of summaries $\theta \in \mathbb{Z}^+$ and the summary size bounds $k, k^- \in \mathbb{Z}^+$ as input, and returns a subset $S$ of $V$ maximizing the density of $H[S]$ for DSD. The algorithm takes the average of estimations obtained through $\theta$ neighborhood summaries for each node in $H$ to achieve good approximation guarantees for peeling coefficients and subgraph densities. It first initializes the empty buffer $\mathcal{K}$ for storing the neighborhood summaries and array $I$ indicating the index of summaries to be constructed and then constructs the $\theta$ summaries (Lines 1–2). Then, the peeling iterations proceed until only one node remains (Lines 3–9). In each iteration, the algorithm selects and removes the node with the smallest peeling coefficient by calling API coefficient (Lines 4–5). Then, the algorithm calls procedure update to perform lazy maintenance of the summaries and obtains the indices $I$ of summaries that need to be reconstructed (Line 6). Then it reconstructs the summaries indicated in $I$ over the current remaining subset $S'$ (Line 7). At the end of each iteration, API density is called to estimate the density of $H[S']$ (Line 8) and the subset that induces larger density is maintained in $S$ (Line 9). Finally, the subset $S$ inducing the largest density obtained during the peeling iterations is returned as the result (Line 10).

---

**Algorithm 1: Summary-based Peeling**

**Input:** KG $\mathcal{G}$, meta-path $\mathcal{M}$, number of summaries $\theta$, summary size bounds $k, k^-$

**Output:** A set of nodes $S \subseteq V$

1  $S \leftarrow \emptyset, \rho \leftarrow 0, S' \leftarrow V, I \leftarrow \{1, 2, 3, ..., \theta\}$ and $\mathcal{K}[\,][\,] \leftarrow \emptyset$;

2  construct( $\mathcal{K}, V, I$ );

3  **while** $|S'| > 1$ **do**

4  $\quad v \leftarrow \arg\min_{u \in S'} \text{coefficient}(u, \mathcal{K})$;

5  $\quad$ remove $v$ from $S'$;

6  $\quad I \leftarrow$ update($\mathcal{K}, v$);

7  $\quad$ construct($\mathcal{K}, S', I$);

8  $\quad \rho' \leftarrow$ density($S', \mathcal{K}$);

9  $\quad$ **if** $\rho' > \rho$ **then** $S \leftarrow S', \rho \leftarrow \rho'$;

10 **return** $S$;

11 **Function** construct($\mathcal{K}, S, I$):

12 $\quad$ **for** $\tau \in I$ **do**

13 $\quad\quad$ **if** $u \in S$ **do** $\mathcal{K}[u][\tau] \leftarrow$ Rand$(0, 1)$;

14 $\quad\quad$ **for** $i = 1$ **to** $L$ **do**

15 $\quad\quad\quad$ **for** $u \in \mathcal{V}_i$ **do** $\mathcal{K}[u][\tau] \leftarrow \oplus_{v \in \mathcal{V}^-(u)} \mathcal{K}[v][\tau]$;

16 $\quad\quad$ **for** $i = L - 1$ **to** $0$ **do**

17 $\quad\quad\quad$ **for** $u \in \mathcal{V}_i$ **do** $\mathcal{K}[u][\tau] \leftarrow \oplus_{v \in \mathcal{V}^+(u)} \mathcal{K}[v][\tau]$;

18 **Function** update($\mathcal{K}, u$):

19 $\quad I \leftarrow \emptyset$;

20 $\quad$ **for** $\tau = 1$ **to** $\theta$ **do**

21 $\quad\quad$ **for** $v \in S'$ **do**

22 $\quad\quad\quad$ remove $r_\tau(v)$ from $\mathcal{K}[v][\tau]$;

23 $\quad\quad\quad$ **if** $|\mathcal{K}[v][\tau]| < k^-$ **then** $I \leftarrow I \cup \{\tau\}$;

24 $\quad$ **return** $I$;

---

### A.4 Theorems and Proofs

**Theorem 1.** *Let $S$ denote the subgraph returned by SANS, we have $\rho_e(S) \geq \frac{\rho_e(S^*)}{2(1+\epsilon)}$ w.p. at least $(1-p)$, provided $\theta = \Theta(\frac{\Lambda}{\epsilon k} \log \frac{|V|}{p})$.*

To prove Theorem 1, we first give the following lemma. We use $S_t$ to denote the subgraph generated at the $t$-th peeling iteration.

**Lemma 1.** *For any $0 \leq t < |V|$, we have $\frac{\rho_e(S_t)}{1+\delta} \leq \widetilde{\rho}_e(S_t) \leq \frac{\rho_e(S_t)}{1-\delta}$ holds w.p. at least $(1-p)$, provided $\theta = \Theta(\frac{\Lambda}{\delta k} \log \frac{|V|}{p})$.*

**Proof Sketch.** We first show that for any $\delta, p \in (0, 1)$ and any node $v \in S_t$, if $\theta = \Theta(\frac{\Lambda}{\delta k} \log \frac{1}{p})$, then $(1 - \delta) \cdot N_{S_t}(v) \leq \widetilde{N}_{S_t}(v) \leq (1 + \delta) \cdot N_{S_t}(v)$ holds w.p. at least $1 - p$. Then, by using the union bound over all nodes in $S_t$, we have $\rho_e(S_t) = \frac{\sum_{u \in S_t} N_{S_t}(u)}{2 \cdot |V_{S_t}|} \geq \frac{\sum_{u \in S_t} \widetilde{N}_{S_t}(u)}{2(1+\delta)|V_{S_t}|} = \frac{\widetilde{\rho}_e(S_t)}{1+\delta}$ and similarly $\rho_e(S_t) \leq \frac{\sum_{u \in S_t} \widetilde{N}_{S_t}(u)}{2(1-\delta)|V_{S_t}|} = \frac{\widetilde{\rho}_e(S_t)}{1-\delta}$ holds simultaneously w.p. $(1-p)$. $\square$

Based on Lemma 1, we give the proof of Theorem 1.

**Proof of Theorem 1.** For a node $v \in S^*$, since $S^*$ is the densest subgraph of $H$, we have $\rho_e(S^*) \geq \rho_e(S^* - \{v\})$, i.e., $\frac{|E_{S^*}|}{|V_{S^*}|} \geq$

$\frac{|E_{S^*}| - N_{S^*}(v)}{|V_{S^*}| - 1}$. Thus,

$$N_{S^*}(v) \geq \frac{|E_{S^*}|}{|V_{S^*}|} = \rho_e(S^*). \tag{7}$$

Consider the first iteration in which a node $u \in S^*$ is going to be removed by Alg. 1 and let $S_t$ denote the subgraph just before $u$ is removed. According to Lemma 1, for any node $v' \in S_t$, we have

$$N_{S_t}(v') \geq \frac{\widetilde{N}_{S_t}(v')}{(1+\delta)} \geq \frac{\widetilde{N}_{S_t}(u)}{(1+\delta)} \geq \frac{(1-\delta)}{(1+\delta)} \cdot N_{S_t}(u) \tag{8}$$

holds w.p. at least $(1 - p)$, given $\theta = \Theta(\frac{\Lambda}{\delta k} \log \frac{1}{p})$. The second inequality in (8) holds since the nodes in $H$ are removed in the ascending order of estimated degrees. Then, by applying union bounds over all nodes in $S_t$, we have

$$\rho_e(S_t) = \frac{\sum_{v' \in S_t} N_{S_t}(v')}{2|V_{S_t}|} \geq \frac{(1-\delta)\sum_{v' \in S_t} N_{S_t}(u)}{2(1+\delta)|V_{S_t}|} \geq \frac{(1-\delta)}{2(1+\delta)}\rho_e(S^*) \tag{9}$$

holds with probability at least $(1 - p)$ provided $\theta = \Theta(\frac{\Lambda}{\delta k} \log \frac{|V|}{p})$. The last inequality holds since $N_{S_t}(u) \geq \rho_e(S^*)$ according to Eq. 7.

Let $S = \arg \max_{0 \leq t \leq |V|} \widetilde{\rho}_e(S_t)$ denote the subgraph returned by Alg. 1. Then according to Lemma 1, we have

$$\rho_e(S) \geq (1-\delta)\widetilde{\rho}_e(S) \geq (1-\delta)\widetilde{\rho}_e(S_t) \geq \frac{1-\delta}{1+\delta}\rho_e(S_t) \geq \frac{(1-\delta)^2}{2(1+\delta)^2}\rho_e(S^*) \tag{10}$$

For any $\epsilon \in (0, 1)$, we can set $\delta = (3 - 2\sqrt{2})\epsilon < 1 + \frac{2-2\sqrt{1+\epsilon}}{\epsilon}$, i.e., $\theta = \Theta(\frac{\Lambda}{\epsilon k} \log \frac{|V|}{p})$. So we have $\frac{1}{2(1+\epsilon)} < \frac{(1-\delta)^2}{2(1+\delta)^2}$ and thus $\rho_e(S) > \frac{\rho_e(S^*)}{2(1+\epsilon)}$ w.p. at least $(1 - p)$. The theorem follows. $\square$

THEOREM 2. *Given any subset $S \subseteq V$ and $v \in S$, we have $\Delta(v) = \mathbb{E}[\tilde{\Delta}(v)]$ and $\rho_\Delta(S) = \mathbb{E}[\tilde{\rho}_\Delta(S)]$.*

PROOF. By dividing $\theta$ neighborhood summaries of node $v$ into 3 partitions and estimating the degree $\tilde{N}(v)$, edge support $\tilde{\Delta}(u,v)$ and sampling adjacent edges of $v$ with different independent partition of neighborhood summaries, we have

$$\mathbb{E}[\tilde{\Delta}(v)] = \frac{\mathbb{E}[\tilde{N}(v)] \cdot \mathbb{E}[\sum_{u \in \mathcal{K}_v} \tilde{\Delta}(u,v)]}{2|\mathcal{K}_v|}$$

$$= \mathbb{E}[\tilde{N}(v)] \cdot \frac{\sum_{u \in \mathcal{N}(v)} \frac{|\mathcal{K}_v|}{N(v)} \mathbb{E}[\tilde{\Delta}(u,v)]}{2|\mathcal{K}_v|} \tag{11}$$

$$= N(v) \cdot \frac{\sum_{u \in \mathcal{N}(v)} \frac{|\mathcal{K}_v|}{N(v)} \Delta(u,v)}{2|\mathcal{K}_v|}$$

$$= \frac{\sum_{u \in \mathcal{N}(v)} \Delta(u,v)}{2} = \Delta(v)$$

The first equality is due to the independence between $\tilde{N}(v)$ and $\sum_{u \in \mathcal{K}_v} \tilde{\Delta}(u,v)$; the second equality is due to the independence of adjacent edge sampling and the estimation of $\tilde{\Delta}(u,v)$; the third equality is due to the unbiasedness of estimation based on the neighborhood summaries according to [4]. Thus, the estimator $\tilde{\Delta}(v)$ is unbiased. The unbiasedness of $\tilde{\rho}_\Delta(S)$ follows trivially from the addition rule of expectation. $\square$

THEOREM 3. *The time complexity of summary-based peeling is $O(k(\theta + \theta^*)(|\mathcal{E}^*| + |V|f(k) + |V|\log|V|))$ and the space complexity*

**Table 3: Time consumption of relational graph materialization based on boolean matrix multiplication (BoolAP, BoolAP$^+$) and Leapfrog TrieJoin.**

| Dataset | BoolAP | BoolAP$^+$ | Leapfrog TrieJoin |
|---------|--------|-----------|-------------------|
| IMDB | 0.00017s | 0.004s | 0.0011s |
| ACM | 19.33s | 0.3s | 3.0014s |
| DBLP | 4.75s | 3.43s | 4.7200s |
| PubMed | 15.36s | 10.9s | 4.4173s |
| FreeBase | 177.04s | 110.7s | 17.4360s |

*is $O(k\theta|\mathcal{V}^*| + |\mathcal{E}^*|)$, where $f(k)$ denotes the time complexity for peeling coefficient estimation using neighborhood summaries and $\theta^*$ denotes the total number of neighborhood summaries reconstructed during the peeling iterations.*

PROOF. **Time complexity.** There are mainly two contributors to the time complexity of the summary-peeling algorithm. First, the algorithm needs to construct the neighborhood summaries. The construction of neighborhood summaries requires propagating at most $k$ random numbers over the matching graph with $|\mathcal{E}^*|$ edges, which takes $O(k \cdot |\mathcal{E}^*|)$ time. The algorithm constructs $\theta$ neighborhood summaries at the beginning of the algorithm and reconstructs $\theta^*$ summaries that are disabled by node removals during peeling iterations. Thus, the time complexity of summary construction is $O((\theta + \theta^*)k \cdot |\mathcal{E}^*|)$. Then, at each peeling iteration, the algorithm updates the neighborhood summaries by removing from $\mathcal{K}$ the random number corresponding to the removed node, re-estimates the peeling coefficients with updated neighborhood summaries, and maintains the min-heap for identifying the node with smallest peeling coefficient. Since $(\theta + \theta^*)$ neighborhood summaries are constructed for each node, at most $O(k|V|(\theta + \theta^*))$ neighborhood summary updates will be performed. For each neighborhood summary update, it will takes $f(k)$ time to re-estimate the peeling coefficient and $O(\log|V|)$ time to maintain the min-heap. The neighborhood summary update itself can be performed in $O(1)$ utilizing an inverted list from random numbers to neighborhood summaries. Thus, the time complexity is $O(k|V|(\theta+\theta^*)(f(k)+\log|V|))$. Therefore, the total time complexity of the summary-peeling algorithm is $O(k(\theta + \theta^*)(|\mathcal{E}^*| + |V|f(k) + |V|\log|V|))$.
**Space complexity.** The firsts term $O(k\theta|\mathcal{V}^*|)$ is the memory consumption of neighborhood summaries and the second term $O(|\mathcal{E}^*|)$ is the memory consumption of the matching graph $\mathcal{G}^*$. $\square$

## A.5 Additional Experiments

**Time consumption of materialization.** Table 3 reports the time consumption of methods BoolAP and BoolAP$^+$ [14] based on boolean matrix multiplication as well as Leapfrog TrieJoin for relational graph materialization across datasets. We can observe that over large datasets PubMed and FreeBase, Leapfrog TrieJoin is significantly faster than BoolAP and BoolAP$^+$. Thus, we adopt Leapfrog TrieJoin to materialize the relational graph for all baselines.
**Scalability.** We test the scalability of methods based on SANS through experiments over large synthetic datasets generated by the TPCH benchmark with scale factors $\lambda \in \{1, 2, 4, 8, 16\}$. The number of nodes in these generated TPCH datasets varies from 1.85M~29.6M,

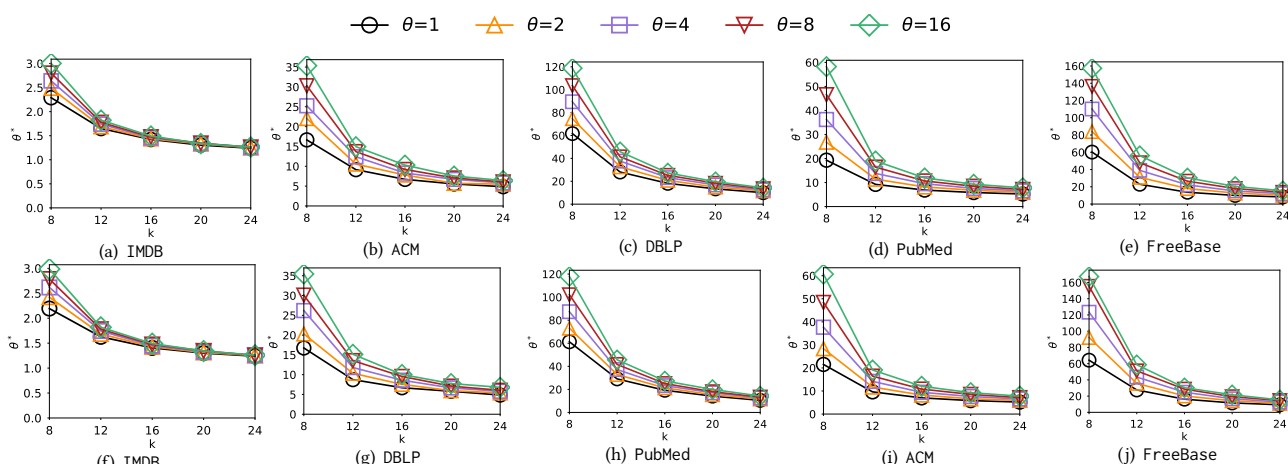

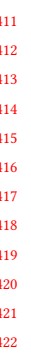

Figure 7: Number of reconstructed summaries during $\theta^*$ during peeling iterations of SansE and SansT varying $k$ and $\theta$. Subfigures (a)-(e) reports the number of reconstructed summaries of SansE; subfigures (f)-(j) reports the number of SansT.

the number of edges varies from 7.5M to 120M. We conduct our experiments with the meta-path (customer)→ (order)→(product), which includes a relational graph connecting pairs of customers who bought the same product [33]. We can observe that both the running time and the memory usage of neighborhood summaries are linear in the size of the dataset, which verifies the scalability of SANS based methods. Specifically, our method SansE and SansT can handle the largest TPCH dataset with ~30M nodes within 260 seconds and 436 seconds respectively with 7.72GB additional memory for neighborhood summaries.



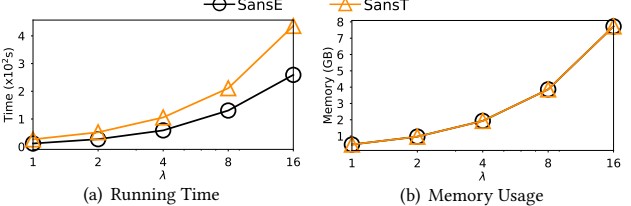

Figure 8: The running time and memory usage of methods SansE and SansT. The $x$ axis uses a log scale.

**Number of reconstructed summaries** $\theta^*$. Fig. 7 reports the number of reconstructed summaries for each dataset during the peeling iterations varying $k$ and $\theta$. We can observe that the number of reconstructed summaries decrease monotonically with the increase of $k$, while increase with the increase of $\theta$.

**Varying** $k^-$. Fig. 9 reports the time consumption and the number of neighborhood summaries reconstructed $\theta^*$ of SansE and SansT varying parameter $k^-$ across datasets. We can observe that both the time consumption of SansE and SansT grows exponentially with the increase of parameter $k^-$ due to the exponential increase of number of summaries reconstructed during the peeling iterations.

**Synthetic Analysis of** $\theta^*$. In addition to reporting the parameter $\theta^*$ for the real-world datasets, we explore possible factors that may influence the number of summary reconstructions $\theta^*$ in our SANS based peeling algorithms. To that end, we perform an analysis of $\theta^*$ over synthetic graphs generated with different models. Specifically, we examine the random binomial graphs and the scale-free graphs generated with the Erdős–Rényi (ER) model and Barabási–Albert

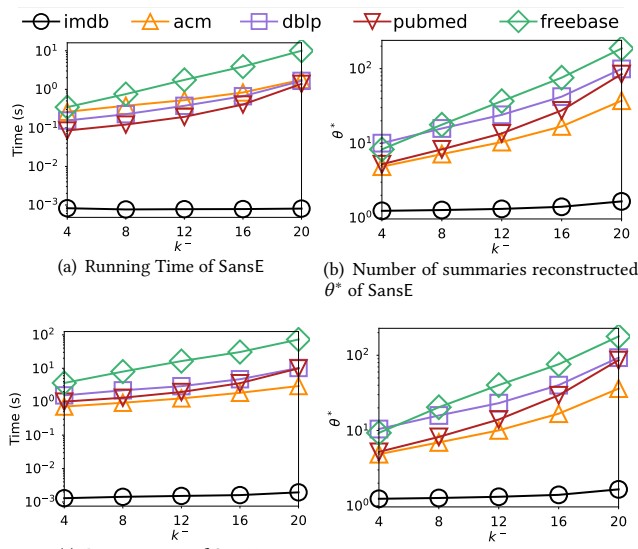

Figure 9: Efficiency Analysis of SansE and SansT varying parameter $k^-$ across datasets. Subfigure (a) and (b) reports the time consumption and the number of summaries reconstructed $\theta^*$ of SansE; subfigure (c) and (d) reports the corresponding evaluation metrics of SansT.

(BA) model respectively. Fig. 10 reports the value of $\theta^*$ while varying the number of nodes $|V|$ in the graph exponentially from 1000 to 16000. The value of $\theta^*$ grows logarithmically with the number of nodes $|V|$ in the synthetic graphs on both binomial graphs (Fig. 10(a) and 10(b)) and scale-free graphs (Fig. 10(c) and 10(d)), for both methods SansE and SansT. Based on this observation, we conjecture that the number of reconstructed summaries required by the peeling algorithm SansE and SansT are logarithmic in the number of nodes in the graph regardless of the specific structure of the relational graphs. We leave a formal proof (or counterexample) of this conjecture as a future work.

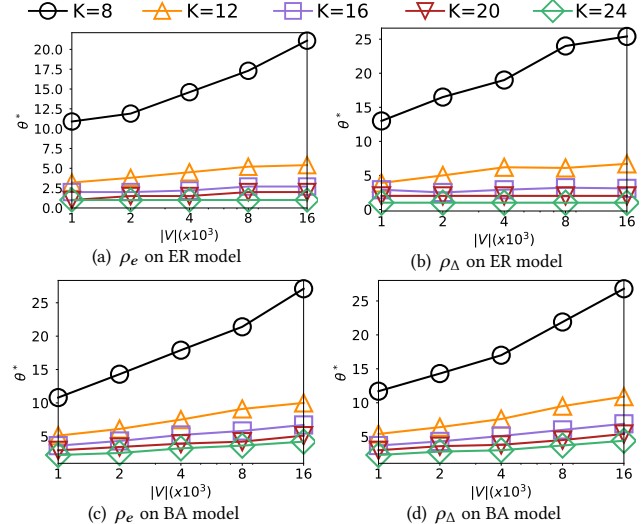

(a) $\rho_e$ on ER model

(b) $\rho_\Delta$ on ER model

(c) $\rho_e$ on BA model

(d) $\rho_\Delta$ on BA model

**Figure 10: Number of summaries reconstructed $\theta^*$ during peeling iteration with varying sizes of synthetic graphs.**

**Table 4: The ratio between the size of densest subgraph discovered by SansE and PeelE.**

| Dataset | IMDB | ACM | DBLP | PubMed | FreeBase |
|---|---|---|---|---|---|
| **Size ratio** | 101.6% | 102.9% | 102.1% | 101.9% | 102.5% |

**Size Comparison.** Table 4 reports the average ratio between the size of densest subgraphs discovered by our method SansE and the materialization-based peeling method PeelE. We can observe that our method SansE reveals densest subgraphs with comparable sizes (consistently less than 3% relative differences across datasets) to the densest subgraphs discovered by PeelE.

