# OpenReview forum: "SANS: Efficient Densest Subgraph Discovery over Relational Graphs without Materialization"
_ACM.org/TheWebConf/2025/Conference — WWW 2025 Poster_

### Official Review · Reviewer_egQe · 2024-11-21

**Novelty:** 5
**Technical Quality:** 5

**Review:**

Strong points:
1.	The paper introduces a novel system, SANS, for efficiently discovering the densest subgraph over relational graphs without materialization. The authors have conducted extensive experiments on multiple real-world datasets to showcase the efficacy and efficiency of the SANS system compared to conventional baselines.
2.	Overall, the topic and the problem of the paper is clearly defined.

Weak points:
1.	Some relatively important content that will affect the reader's understanding should be placed in the main body, not in the appendix. Moreover, some sections may require more explanation or restructuring to ensure that all readers can understand the theoretical analysis.
2.	The writing and organization of the paper are generally clear, but further proofreading may be necessary to eliminate any grammatical or spelling errors. The absence of definite and indefinite articles exists in the paper. There are also some grammatical errors, like line 455 should be “To avoid reconstructing”.

**Questions:**

1.	It is a pity that the innovation of this paper is not well expressed in the paper, and the algorithm elaboration and case analysis account for a relatively small proportion.
2.	The paper could discuss potential directions for future research, such as how to further optimize the SANS system or how to apply it to other types of graph analysis problems.

**Reviewer Confidence:**

3: The reviewer is confident but not certain that the evaluation is correct

**Scope:**

3: The work is somewhat relevant to the Web and to the track, and is of narrow interest to a sub-community

---

### Official Review · Reviewer_Um29 · 2024-11-22

**Novelty:** 5
**Technical Quality:** 4

**Review:**

This paper introduces SANS, an efficient system for dense subgraph discovery (DSD) that eliminates the need for resource-intensive graph materialization. SANS employs a summary-based peeling method for estimating subgraph densities, demonstrating orders of magnitude efficiency over materialization-based methods.

Theoretical analysis shows SANS provides a (2+$\epsilon$)-approximation for edge-density metrics.  Experiments are also conducted to evaluate its efficiency and effectiveness, demonstrating significant speedups and maintaining high accuracy compared to traditional materialization-based approaches.

Pros

1. SANS eliminates the need for graph materialization, significantly reducing computational costs.

2. The inclusion of user-defined APIs allows users to customize the system based on their specific needs, making SANS adaptable to a wide range of applications.

3. SANS is also tested on heterogeneous data obtained from real-world applications.

4. This paper presents the notations and problem formulation before discussing the related works, allowing readers to gain a better understanding of the context.

Cons

1. The paper only proves the approximation ratio for edge density metric, lacking a more general approximation proof for other density metrics.

2. The evaluation of SANS’s effectiveness is limited. When evaluating the effectiveness of SANS, the article only compared it with a single baseline.

**Questions:**

1. The paper only proves the approximation ratio for edge density. How can the performance of SANS be ensured when using other density metrics?

2. When applying the SANS system on different datasets or real-world applications, is there a more effective approach to determine the optimal hyperparameters (e.g., $k$ and $\theta$) rather than trial-and-error experiments?

**Reviewer Confidence:**

2: The reviewer is willing to defend the evaluation, but it is likely that the reviewer did not understand parts of the paper

**Scope:**

4: The work is relevant to the Web and to the track, and is of broad interest to the community

---

### Official Review · Reviewer_UWnm · 2024-11-27

**Novelty:** 5
**Technical Quality:** 5

**Review:**

The paper introduces SANS, a system for efficiently discovering the densest subgraph in relational graphs without the need for graph materialization, using a summary-based peeling approach and neighborhood summaries to estimate peeling coefficients and subgraph densities.

Pros:
- The paper presents a novel approach to densest subgraph discovery that significantly reduces the computational overhead by avoiding graph materialization, which is a practical advantage for handling large-scale datasets.
- SANS demonstrates flexibility in supporting various density metrics, making it a versatile tool for different types of relational graph analysis.
- The experimental results show that SANS achieves high efficiency and accuracy, outperforming conventional methods in both speed and accuracy metrics.

Cons:
- How does the SANS system ensure the quality of density estimation when the size of the neighborhood summaries is small?
- Can the authors elaborate on the scalability of SANS when applied to extremely large graphs with millions of nodes and edges?
- The paper could benefit from a deeper analysis of the trade-offs between the accuracy of density estimation and the computational efficiency of the SANS system.
- How does SANS handle the selection of the meta-path in the heterogeneous data source?
- Could the authors provide more details on how the summary-based peeling algorithm scales with the increase in the number of nodes and edges in the graph, particularly for very large-scale graphs?

**Questions:**

See Cons.

**Reviewer Confidence:**

4: The reviewer is certain that the evaluation is correct and very familiar with the relevant literature

**Scope:**

4: The work is relevant to the Web and to the track, and is of broad interest to the community

---

### Official Review · Reviewer_b6QV · 2024-11-29

**Novelty:** 4
**Technical Quality:** 3

**Review:**

Pros
1. The proposed SANS system introduces a novel materialization-free approach for densest subgraph discovery over relational graphs. This significantly reduces computational overhead compared to traditional methods relying on materialized graphs.

2. The paper benchmarks SANS against several state-of-the-art algorithms, demonstrating its advantages in both efficiency and scalability across multiple datasets and density metrics.

Cons:
1. The efficiency of SANS relies heavily on the quality of the neighborhood summaries. A deeper exploration of how graph sparsity or connectivity impacts performance would be valuable.

2. Although the paper provides high-level theoretical results for time and space complexity, the analysis lacks more detailed insights into the algorithm’s performance under varying graph structures or edge densities.

**Questions:**

1. Could you provide more detailed guidelines for selecting k and θ across diverse datasets?
2. How does SANS perform in scenarios where the underlying relational graph is sparse?

**Reviewer Confidence:**

3: The reviewer is confident but not certain that the evaluation is correct

**Scope:**

3: The work is somewhat relevant to the Web and to the track, and is of narrow interest to a sub-community

---

### Official Review · Reviewer_gnBb · 2024-12-03

**Novelty:** 5
**Technical Quality:** 6

**Review:**

This submission is well-written and well-structured, making it accessible to readers who may not be familiar with the densest subgraph discovery (DSD) problem. The examples provided on pages 2 and 4 are particularly helpful in illustrating the concept.

I understand that materializing relational graphs can be resource-intensive, which highlights the significance of the contribution made by avoiding this materialization while discovering the densest subgraph.

Although I am not an expert in the data management field and have limited knowledge of DSD research, I believe the contribution of this submission is strong, and I am inclined to accept it.

Minors
- I recommend that the authors include additional discussions regarding the connections between the DSD problem and the Web community.
- Typo corrected: line 121, "it."

**Questions:**

Please see above.

**Reviewer Confidence:**

1: The reviewer's evaluation is an educated guess

**Scope:**

2: The connection to the Web is incidental, e.g., use of Web data or API